# Detection of Mosaic Absence of Heterozygosity (AOH) Using Low-Pass Whole Genome Sequencing in Prenatal Diagnosis: A Preliminary Report

**DOI:** 10.3390/diagnostics13182895

**Published:** 2023-09-09

**Authors:** Yan Lü, Yulin Jiang, Xiya Zhou, Na Hao, Chenlu Xu, Ruidong Guo, Jiazhen Chang, Mengmeng Li, Hanzhe Zhang, Jing Zhou, Wei (Victor) Zhang, Qingwei Qi

**Affiliations:** 1Department of Obstetrics and Gynecology, National Clinical Research Center for Obstetric & Gynecologic Diseases, Peking Union Medical College Hospital, Chinese Academy of Medical Science & Peking Union Medical College, Beijing 100730, China; 2AmCare Genomics Lab, Guangzhou 510335, Chinavictor.w.zhang@amcarelab.com.cn (W.Z.)

**Keywords:** absence of heterozygosity, mosaicism, whole genome sequencing, prenatal diagnosis

## Abstract

**Objective**: Mosaicism is a common biological phenomenon in organisms and has been reported in many types of chromosome abnormalities, including the absence of heterozygosity (AOH). Due to the detection limitations of the sequencing approach, mosaic AOH events are rarely assessed in clinical cases. Herein, we report the performance of mosaic AOH identification using a low-pass (5~8-fold) WGS method (termed ‘CMA-seq’, an abbreviation for ‘Chromosome Analysis by Sequencing’) in fetal genetic diagnosis. **Methods:** Thirty AOH-negative, eleven constitutional AOH, and three mosaic AOH samples were collected as training data sets to develop the algorithm and evaluate the suitable thresholds for distinguishing mosaic AOH. Twenty-four new chromosomal aberrant cases, along with sixteen constitutional AOH samples, which were previously ascertained via the SNP-array-based method, were used as a validation data set to measure the performance in terms of sensitivity and specificity of this algorithm. **Results:** A new statistic, ‘D-value’, was implemented to identify and distinguish constitutional and mosaic AOH events. The reporting thresholds for constitutional and mosaic AOH were also established. In the validation set consisting of 24 new cases, seven constitutional AOH cases and 1 mosaic AOH case were successfully identified, indicating that the results were consistent with those of the SNP-array-based method. The results of all sixteen constitutional AOH validation samples also met the threshold requirements. **Conclusions:** In this study, we developed a new bioinformatic algorithm to accurately distinguish mosaic AOH from constitutional AOH by low-pass WGS. However, due to the small sample size of the training data set, the algorithm proposed in this manuscript still needs further refinements.

## 1. Introduction

Mosaicism, which is conceptually different from chimerism, refers to the abnormal situation in which an individual has developed from a single fertilized egg and has two or more populations of cells with distinct genotypes [1]. It has been reported for many types of chromosome abnormalities such as polyploidy, aneuploidy, rings, copy number variant (CNV), and others, including the absence of heterozygosity (AOH), which has not been relatively fully recorded [2].

In fact, compared with a constitutional mutation, mosaicism, as a state of better tolerated or life-compatible, is more common and has pervasiveness in normal individuals. For example, the AOH phenomenon was first discovered as a mosaic state in *Drosophila* by Stern in 1936 [3], which is one of the earliest descriptions of mosaicism, far earlier than the concept of constitutional uniparental disomy (UPD), with the latter first proposed in 1980 by Eric Engel [4], who reported that it was likely to occur through meiotic error, and was demonstrated as a mechanism for human genetic disorder in 1988 [5].

A mutation accumulation study in yeast has demonstrated that the rate of mosaic AOH is much higher than the rate of point mutations [6] and has greater functional potential due to their larger size. Although mosaicism typically shows a milder phenotype and is better tolerated than a constitutional mutation, the detection of mosaic AOH has important clinical significance and poses exceptional challenges in the scenario of prenatal testing and genetic testing of products of conception. For example, mosaicism for paternal UPD of the chromosomal segment 11p15 → pter is found in approximately 10–20% of cases with Beckwith–Wiedemann syndrome [7]. Another typical example is the mosaic UPD of chromosome 15, which may partially explain the significant difference in the proportion of UPD pathogenicity (2% vs. 25%) between the Angelman syndrome and Prader–Willi syndrome [8]. In addition, mosaic AOH is genetically correlated with age-dependent complex traits such as atherosclerosis, Alzheimer’s disease, non-insulin-dependent diabetes, and neoplastic disorders [9].

The specific AOH subtype and characteristics reflect their distinct underlying formation mechanisms. Constitutional AOH is associated with meiotic errors, whereas mosaic AOH mainly occurs as a result of a mitotic error in somatic cells. Specifically, constitutional AOH is caused primarily by meiotic errors (nondisjunction in meiosis I or II, or anaphase lag), followed by trisomy rescue/monosomy rescue, or rarely, by gamete complementation, or simply caused by parental consanguinity. Mosaic AOH can arise from mitotic nondisjunction/anaphase lag, leading to entire chromosome mosaic AOH [9]. Another formation mechanism, double-strand breaks and subsequent recombination/break-induced replication is responsible for interstitial mosaic AOH, while the major subtype, terminal mosaic AOH, is caused by mitotic crossing over [10]. The whole genome mosaic AOH involves a failure of replication of the maternal genome prior to the first cleavage, with normal replication and segregation of the paternal genome [11]. Gene conversion, resulting in short stretches of homozygosity (typically less than 10 kb) [12,13], is not the scope of our discussion.

Many monogenic and genomic disorders are detectable in the prenatal period, and the capacity to identify them has increased remarkably as molecular testing techniques continue to improve and become incorporated into clinical practice. Chromosomal microarray analysis (CMA) has been recommended as the first-tier diagnostic test since 2013 [14], but the limitations are obvious. Array comparative genomic hybridization (aCGH) cannot detect triploids or AOH, while single nucleotide polymorphism (SNP) array, due to the limited and uneven distribution of the probes, as well as the discrepancy of the cutoff values for reporting between clinical laboratories, often leads to results discrepancies [15]. Meanwhile, as the cost and platform versatility of CMA are of great concern, next-generation sequencing (NGS) has become a mainstream platform, benefiting from continuously decreasing sequencing costs.

Short-read NGS is transforming prenatal genetic testing at an unprecedented pace. Over the past few years, the versatility of NGS has made the detection of AOH feasible [16]. Although high-depth whole genome sequencing (WGS) holds the promise of the most accurate AOH detection, the current high cost inhibits widespread clinical implementation. Previous studies [17,18] showed the feasibility of applying low-pass WGS in the identification of clinically significant AOHs; however, for detecting mosaic AOH, they only described broad principles, and no specific algorithm was provided. In our previous publication [19], we introduced ‘CMA-seq’ (a method of 5~8-fold short-read WGS) showing equivalent AOH detection power to that of routine CMA (750K), yet the performance of mosaic AOH detection was not considered. Currently, there are only a few such investigations because the pipeline for detecting AOH entails following a typical variant calling pipeline and then analyzing allele frequencies, whereas low-pass WGS is unable to accurately calculate the variant allele fraction (VAF), which results in a high likelihood of incorrect genotyping. Herein, as a solution, we present the performance of mosaic AOH detection using CMA-seq technology. Specifically, we initially used common single-nucleotide variants (SNVs) for sample genotyping and utilized the VAF value to classify all SNVs into four types of genotype status, namely AB, BB, ABB, and AAB. We hypothesized that the amounts of ABB allele and AB allele within a given genomic region were suitable for identifying mosaic AOH events. By training on a set of known AOH samples, a new statistic (D-value) was introduced, and thresholds were established to support the detection of mosaic AOH in prenatal samples. Finally, the performance of the newly developed algorithm was assessed in an independent validation cohort.

## 2. Materials and Methods

### 2.1. Study Design and Sample Preparation

SNVs are regions that have two different alleles at the same locus on different homologs. The genotype status is used to determine whether a region has undergone AOH event to become homozygous. As we implicitly assumed in the previous study [19], normal individuals are expected to possess a quantity of heterozygous alleles for a given genomic region. If not, and concomitant with an abnormal increase in homozygous variants, then this region is considered as putative constitutional AOH. In this study, we further hypothesized that the amount of ABB allele (defined as SNV with VAF between 0.65 and 1; see Section 2.3.2 for details) in normal individuals should not be high, if higher than expected, and coupled with diminishing heterozygous alleles, mosaic AOH should be suspected (Figure 1). To test this simple model, we collected AOH-negative, constitutional, and mosaic AOH samples as the training data set to find suitable statistics and determine optimal thresholds and resolution for reporting potential mosaic AOH events. Twenty-four new clinical cases and sixteen constitutional AOH samples were used as the validation data set to assess the reliability of this method and evaluate the clinical sensitivity and specificity of the novel bioinformatic algorithm (Figure 2). This study was approved by the Institutional Review Board of the Peking Union Medical College Hospital (IRB no. S/K 13023670). 

### 2.2. Samples

The benchmark cohort, which consisted of 28 unrelated individuals, was used to generate crucial benchmark values for the Z_score calculation (Appendix A). The training data set included 30 AOH-negative samples (Appendix A), 11 constitutional AOH samples (Appendix A), and 3 clinical mosaic AOH samples (Appendix A). AOH-negative samples were obtained from phenotypically normal individuals, including 13 females and 17 males, with a mean age of 33.5 years. The validation data set included 24 chromosomal aberrant prenatal cases (Appendix A) and 16 constitutional AOH samples (Appendix A). All samples in the benchmark cohort, training, and validation data sets, except for AOH-negative samples, had been previously assessed by CMA (750 K). The negative samples were not subjected to CMA testing because the carrier rate of large-sized (>4 Mb) AOH in the normal adult population is considerably low [20,21].

### 2.3. Methods

#### 2.3.1. Sequencing, Quality Control, and Variant Calling 

Genomic DNA from the benchmark cohort, training, and validation data sets was extracted using the standard operating instructions of the QIAamp DNA Blood Mini Kit (Qiagen, Valencia, CA, USA), followed by random fragmentation with the Q800R Sonicator (Qsonica, Newtown, CT, USA). Library construction and sequencing on the AmCareSeq-2000 sequencer (AmCare Genomics Lab, Ltd., Guangzhou, China) was conducted according to the instructions, with a 200~500 bp insert size and PE150 sequencing strategy. 

The sequencing depth was 5~8X, with a total number of raw reads >100 M for each single sample. Reads with a base quality of less than QC20 were discarded. The quality control metrics were as follows: Q30 ≥ 85%, average sequencing coverage ≥5X, and count of CNVs with size larger than 100 kb ≤ 20. Detecting and estimating DNA sample contamination were conducted by VerifyBamID2 [22].

After QC assessment, high-quality reads were mapped to the reference human genome version GRCh37/hg19 by Burrows–Wheeler Aligner (BWA) [23]. SNVs were called by an in-house bioinformatics pipeline [24]. Any SNVs with read depth < 5, or >arg + 4 s.d. (>4 s.d. from the mean) fold were excluded from our analysis. The reason for excluding SNVs with particularly high or low coverage is to guarantee the accuracy of downstream calculations. The removal of loci of low depth was due to a sharp decrease in the accuracy of variant calling when the sequencing depth was below 5X. Removing loci with high depth was because they were likely artifacts derived from the high CG or repetitive genomic regions, where read alignment accuracy was low and variant calling was largely inaccurate. Removing these loci can avoid introducing irrelevant confounding factors in downstream analysis.

#### 2.3.2. Genotype Status

SNVs were classified into four distinct categories based on the VAF value: AB (0.35 < VAF < 0.65), BB (VAF = 1), ABB (0.65 < VAF < 1), and AAB (VAF < 0.35). VAF was calculated as the number of reads supporting the mutant base type divided by the total number of reads at the locus. Note that the AA allele was not mentioned because they were not called on the VCFs (variant call files) as variants; the BB allele is abbreviated as the B allele hereafter. The influence of sequencing depth on genotyping can be found in Appendix A.

#### 2.3.3. Bioinformatics Analysis

Estimation of Benchmark Data

Defining the bin of 200 kb in size as the minimum statistical unit (the reason for selecting 200 kb as the optimum bin size can be found in reference [19]). The count of each genotype status (i.e., AB, B, ABB, and AAB) was calculated individually for every bin using selected SNVs with read depth between 5-fold and (arg+4s.d.)-fold, and the mean (bm_bin_avg) and standard deviation (bm_bin_s.d.) values of the count were calculated for all 28 benchmark samples for each genotype status of each such non-overlapping bins.

Calculation of the Z_Score and D-Value of Bin

Any SNVs with read depth < 5, or >(arg+4s.d.)-fold were excluded from the test sample analysis, where ‘arg+4s.d.’ refers to the value obtained from 28 benchmark samples. 

The count and Z_score of each genotype status within the 200 kb range were calculated. Specifically, the calculations were as follows:AB_count_Z_score = (AB_count − AB_bm_bin_avg)/AB_bm_bin_s.d.
B_count_Z_score = (B_count − B_bm_bin_avg)/B_bm_bin_s.d.
ABB_count_Z_score = (ABB_count − ABB_bm_bin_avg)/ABB_bm_bin_s.d.
where the bm_bin_avg and bm_bin_s.d. of each genotype status were obtained from benchmark samples; for a given bin, it is a fixed value. 

Two new statistics, D1 = |B_count_ Z_score _arg-AB_count_ Z_score _arg|, and D2 = |ABB_count_ Z_score _arg-AB_count_ Z_score _arg|, were introduced; note that the theoretical minimum statistical scope for D-value was bin. 

Calculation of Z_Score and D-Value of Windows

Scan each autosome by moving a window of fixed size along its length in search of stretches of consecutive homozygous variants. A 2 Mb (i.e.,10 consecutive 200 kb bins) size was taken as a sliding window, the stepping size was 1 bin, and the Z score and D-value were calculated for every possible 2 Mb sliding window.

Sliding Windows and Candidate AOH Identification

If the D1 and D2 values for a given bin met a defined threshold, they were marked as putative constitutional/mosaic AOH. Merging two adjacent blocks with intervals of less than 1 Mb was performed automatically. If the final size of the merged blocks was smaller than 4 Mb, it was filtered out.

Boundary Refinement

AOH with size >4 Mb were marked for further manual examination for refinement of the precise AOH boundaries. We defined the left boundary as ‘L’ and the right boundary as ‘R’. For the marked candidate AOH region, we attempted to determine if the extended region from (L-1 Mb) to R satisfied the cutoff. If satisfied, the region was replaced by ‘(L-1) to R’, and iteration was performed. Following a similar logic, attempting to determine if the extended region from the new L to (R + 1) satisfied the cutoff; if satisfied, replace the region with L to R + 1, continuously iterating until the maximum region that met the cutoff is found. Figure 3 illustrated the algorithm and procedure of the bioinformatics analysis.

#### 2.3.4. Chromosomal Microarray Analysis

The Affymetrix^®^ CytoScanTM 750 K Array (Affymetrix, Santa Clara, CA, USA) was used for AOH analyses for all samples in the benchmark cohort, training, and validation data sets, except for AOH-negative samples. See [19] for details. The threshold for reporting autosomal AOH is ≥5 Mb for terminal AOH and 10 Mb for interstitial AOH. The AOH detection calls from CMA and CMA-seq were generated independently and blinded to each other.

## 3. Results

### 3.1. AOH-Negative Samples

We first assessed a total of 30 AOH-negative samples derived from clinically normal individuals to evaluate background levels using CMA-seq. Three randomly selected autosomes (chr1; chr7; chr15) for all samples were examined for a Z_score average dot-plot and D-value. 

As an example, Figure 4 shows the Z_score average dot-plot of chr1 of four randomly selected AOH-negative samples (Neg-1, Neg-13, Neg-19, and Neg-29). The count_Zscore for each genotype status (y-axis) was plotted against the genomic location (x-axis). Each dot represents an average Z_score of one bin (size = 200 Kb). The results showed that the negative samples exhibited roughly similar curve tendencies, i.e., the curve of the AB count_Zscore (red) and the curve of the B count _Zscore (green) almost coincided (for the sake of brevity, this type of graph is named the ‘D1 plot’ hereafter), the curve of the ABB count_Zscore (orange) and the curve of the AB count_Zscore (red) were close to coinciding (this type of graph is named the ‘D2 plot’ hereafter). No gap existed within the two curves of the same plot. 

All samples were found to comply with the following rules (Appendix A, Figure 5): D1 < 0.5 and D2 < 0.6.

### 3.2. Constitutional AOH Samples

Constitutional AOH represents germline abnormalities that are present in all cells. All 11 constitutional AOH samples previously confirmed by CMA were categorized into 3 subgroups, namely whole genome AOH (1 case), whole chromosome AOH (2 cases), and segmental AOH (8 cases) (Appendix A). 

The CMA-seq results showed that case #59 had a whole genome AOH, and all autosomes exhibited the features of D1 > 2 and D2 > 2 (Figure 6A,B). Detailed clinical data for case #59 can be found in reference [19].

Case #58 had a whole chromosome 6 AOH, with D1 value of 3.53 and D2 value of 0.9 (Figure 6C,D). Importantly, the two cases (#58 and #32) with whole chromosome AOH all met the preset criteria of D1 > 2 and D2 < 1.

Case #53 had two segmental AOHs on chr4 ranging in size from 31.1 Mb to 2.3 Mb. The larger AOH region at chr4: 85,400,000–116,500,000 had a D1 value of 2.73 and a D2 value of 0.57, and the smaller AOH region at chr4: 138,700,000–141,000,000 had a D1 value of 2.94 and a D2 value of 0.83. In contrast, the AOH-negative region had a D1 value of 0.14 and a D2 value of 0.39 (Figure 6E,F). All eight segmental AOH cases met the cutoff of D1 > 1.8, D2 < 1 for the AOH region, and D1 < 0.5, D2 < 0.6 for the AOH-negative region (Appendix A). 

### 3.3. Mosaic AOH Cases

Mosaic AOH is defined as AOH present in only a proportion of cells. Three mosaic AOH cases (Cases #28, #60, and #63) previously ascertained by CMA were used to survey mosaic AOH features. As shown in the log2 Ratio figure of the three cases (Appendix A), the possibility of CNV, aneuploidy, and polyploidy was first ruled out by CMA-seq.

#### 3.3.1. Whole Genome Mosaic AOH (Case #28)

A 27-year-old pregnant woman underwent amniocentesis for severe fetal growth restriction. A whole genome mosaic AOH (mosaicism level ~40%) was detected by CMA (Figure 7A, chr1 as a representative), which was speculated to be relevant to the prenatal phenotype. The parents opted to terminate the pregnancy.

As shown in Figure 7C, most autosomes from Case #28 were successfully judged as mosaic AOH (D2 > 0.85 and D1 < 0.5) by CMA-seq. However, two smaller chromosomes (chr20 and chr21) were misinterpreted as AOH-negative (D2 < 0.6), six chromosomes (chr1, 2, 4, 18, 19, and 22) exhibited ambiguous D2 values (0.6 < D2 < 0.85), indicating that the identification for this mosaic type still lacks sensitivity (Appendix A). Figure 7B,D show the D1 plot and D2 plot of chr9.

#### 3.3.2. Whole Chromosome Mosaic AOH (Case #60)

A 29-year-old pregnant woman underwent amniocentesis for a high risk of trisomy 7 (Z = 8.07) indicated by noninvasive prenatal testing (NIPT). CMA testing found no pathogenic CNV but a mosaic (~40%) AOH of whole chromosome 4 (Figure 8A). Fetal anatomic ultrasound found no anomalies. The child developed normally when seen at the age of 4.

Figure 8B,C reveals the results of chr4 by CMA-seq (D1 = 0.79 and D2 = 1.54).

#### 3.3.3. Segmental Mosaic AOH (Case #63)

A 36-year-old pregnant woman underwent amniocentesis for a suspected atrioventricular septal defect of the fetus. A mosaic (~30%) terminal 15.7 Mb AOH at 6p25.3p22.3 (arr[hg19] chr6:203,877–15,972,341) was detected by CMA (Figure 9A). This region encompasses 14 genes associated with autosomal recessive diseases and 2 imprinted genes unrelated to any Mendelian disease. Short tandem repeat testing showed that fetal chromosome 6 was derived from both parents, indicating that UPD was not involved. Trio-WES (Whole Exome Sequencing) found no clinically relevant variants, ruling out potential recessive disorders. Further ultrasound assessment did not detect any cardiac anomalies. The child developed normally when seen at age 3.

The CMA-seq results clearly showed that the ABB_count_Z_score (orange curve) rose at the p-arm telomere of chr6, while the AB_count_Z_score (red curve) slightly dropped at the same region (Figure 9D), resulting in a two curves departure from each other (Figure 9C); the D2 value of the AOH region was 1.18, while the D2 value of AOH-negative region was 0.31 (Appendix A).

### 3.4. Determination of Thresholds and Detection Pipeline

Based on the results above, the thresholds for mosaic AOH were established (Table 1, Figure 10A). For whole chromosome/segmental AOH to be D2 > 1, D1< 1.5, and segmental size > 10 Mb. In addition, the following points need to be noted: (1) ABB_count_Z_score must be a positive value (Appendix A); (2) the current thresholds are suboptimal for smaller chromosomes and pericentromeric AOH, which should be interpreted with caution; (3) the thresholds are currently not applicable to sex chromosomes.

Similarly, thresholds for constitutional AOH were D1 > 2 (stringent) or 1.5 (loose), D2 < 1, and segmental size > 4 Mb.

Note that the thresholds for whole genome AOH, whether mosaic or constitutional, are currently uncertain. 

### 3.5. Validation

To examine the detection power of D-value and the rationality of reporting thresholds, we tested 24 new prenatal cases, which were previously assessed by CMA, using CMA-seq. Ten constitutional AOHs were detected from seven cases by CMA-seq (Table 2, Appendix A). Eight of them were consistent with the previous CMA results, and two were additional detections. Multiple AOHs were detected by CMA-seq in three cases (#164, #166, and #171). There were two novel AOH calls made by CMA-seq but missed by CMA. Inspection of the original data found that CMA also detected the corresponding AOH regions, but they did not meet the reporting cutoff values of CMA. Notably, one AOH event (#170) was detected by CMA but missed by CMA-seq due to the lower D1 value (1.32<1.5) of the corresponding AOH region (Appendix A). Note that the sex chromosomal abnormalities were not analyzed.

Only one case (#167) was identified as mosaic AOH, in line with the results of CMA (Table 2). Here is a detailed description of this complex case.

A 33-year-old pregnant woman underwent amniocentesis for a high risk of trisomy 15 (Z = 4.84) detected by NIPT. CMA found no pathogenic CNV or aneuploidy in amniotic fluid, but a large 79.5 Mb constitutional AOH at 15q11.2q26.3 containing the imprinted region for Prader–Willi/Angelman syndrome. Since CMA has no probe in the p-arms of the acrocentric chromosome 15, the fetus may possess an entire chr15 AOH. Karyotyping found a homologous Robertsonian translocation of chr15. Parental validation by CMA demonstrated that both arms were derived from the maternal chromosome, leading to the confirmed molecular diagnosis of Prader–Willi syndrome. Fetal anatomic ultrasound at 23 weeks of gestation revealed a single umbilical artery without other anomalies. The parents opted to terminate the pregnancy after counseling. 

CMA using the abortive placenta revealed low-level (11%) mosaic trisomy 15 and high-level (89%) AOH at 15q11.2q26.3 (79.5 Mb, Figure 11A), suggestive of placental/fetal discordance. Nondisjunction or anaphase lag in meiosis and incomplete trisomy rescue may be the genetic basis for mosaic AOH formation.

CMA-seq data derived from the abortive placenta revealed that the dosage of chr15 was ×2.47 (Figure 11D), suggesting the presence of mosaic trisomy 15. According to the detection pipeline shown in Figure 10B, the occurrence of mosaic AOH events should be taken into consideration in the presence of mosaic aneuploidy or CNV [25]. This case was finally diagnosed as mosaic trisomy in conjunction with mosaic AOH of chr15 according to the D-value (D1 = 1.72, D2 = 2.29) based on abortive placenta.

Using the 24 prenatal cases, the clinical specificity and sensitivity of our AOH detection algorithm can be preliminarily evaluated. The clinical specificity [true negatives/(true negatives + false positives)] was 100%, and the clinical sensitivity [true positives/(true positives + false negatives)] was 88.9%.

Furthermore, a verification assay was performed on sixteen constitutional AOH samples previously ascertained by CMA. In brief, all samples met the threshold setting, confirming that the threshold setting is reasonable (Appendix A).

## 4. Discussion

Due to limitations in the detection power utilizing the low-pass WGS approach, AOH mosaicism is still rarely detected or involved in disease association studies in clinical practice, and the true incidence of mosaic AOH may be underestimated. However, mosaic AOH detection has important clinical significance, especially for prenatal genetic diagnosis. Early embryogenesis appears to be characterized by remarkably high levels of chromosome abnormalities, where AOH detection is highly recommended. There are differences between constitutional and mosaic AOH in terms of (i) impact on embryonic development, (ii) co-occurrence of mosaic trisomy and mosaic AOH, and (iii) potential recurrence risks [26]. The routine identification and assessment of the clinical significance of mosaic AOH in prenatal diagnosis can provide underlying etiology for a portion of undiagnosed cases. Potential benefits to identifying mosaic AOH also include more accurate genetic counseling about the patient’s prognosis and recurrence risk, as well as guiding therapeutic decisions.

Accurate detection of clinically relevant mosaic variants from sequencing data is challenging [27], especially for AOH, since its formation mechanism per se is complex and unique. Generally, constitutional mutation (arising pre-zygotically) and mosaic mutation (mainly arising post-zygotically) take the timing of mutation events as a barrier [25], but AOH seems to have obscured this rule since the formation of constitutional AOH involves an initial meiotic error and a secondary mitotic correction step (trisomy rescue/monosomy rescue) [28]. Of course, gametic completion is an exception, as this mechanism does not involve post-zygotic events [25]. 

In this study, we developed a new bioinformatic algorithm to accurately detect and score mosaic AOH and optimized the existing method for constitutional AOH identification. In the process of seeking statistics appropriate for mosaic AOH identification, we did not directly utilize extant variables but redesigned new statistics D2 inspired by the seesaw effect in the count of ABB alleles and AB alleles. This strategy is different from current methods [18]. One key improvement of AOH detection is the enhanced detection criteria for constitutional AOH. Previously, merely the AB allele and the B allele (i.e., two components of D1) were considered when confirming constitutional AOH, without implicating D2 [19]. Now, it is essential to consider both D1 and D2 jointly, an important optimization step. Note that this method is not designed for UPD and cannot detect heterodisomy UPD without parental assessment.

To cope with the problem of inaccurate genotyping and calculation of VAF inherent to low-pass WGS [29], this study follows the ideas of scale upgrade of our first article [19], i.e., using 200 kb as the minimum statistical unit, calculating the VAF of four genotype status within the bin scope to reduce the effect of low-pass on genotyping (for example, avoiding the ‘allele dropout’ error from situation that only one of the two chromosomes has been sampled at a specific site) and VAF calculation. 

In the validation process, one constitutional AOH event was detected by CMA but missed by CMA-seq as a consequence of the D1 value of the corresponding AOH region that was 1.32, which was lower than the default threshold of 1.5. Next, we will upgrade the algorithm by introducing weights based on read-depth information to improve the sensitivity of AOH detection.

There are several potential limitations to note in our study. First and foremost, cases of mosaic AOH enrolled in our training data set were scarce (three cases), making it challenging to gain deeper insight into the profile of ABB and AB alleles fluctuation. Second, there were relatively few whole-genome AOH cases, which, whether constitutional or mosaic, exhibited distinctive features compared to whole chromosomal or segmental AOH cases, leading to the whole genome AOH threshold not being determined. Third, for prenatal cases, the mixture of maternal cells with fetal cells would lead to an appearance of mosaic abnormality, and the possibility of maternal cell contamination was not completely excluded in our cases. Fourth, both genotype status and thresholds may be influenced by the genomic context of the AOH (chromosome-specific bias), implying that a larger training set is required. As the clinical sample collection increases, the criteria may need to be further adjusted. Finally, the calculation of the mosaic ratio, AOH of sex chromosomes, and low level of mosaicism was not addressed.

Finally, we emphasize that AOH is a copy-neutral abnormality, and caution is needed in the detection prioritization between AOH and other forms of chromosome abnormalities. The risk of CNV, aneuploidy, and polyploidy must first be ruled out, after which AOH identification is performed. This concept is crucial; otherwise, it will result in spurious AOH identification. However, the possibility of co-occurrence of mosaic trisomy and AOH (AOH in the disomic cells) should not be overlooked.

## Figures and Tables

**Figure 1 diagnostics-13-02895-f001:**
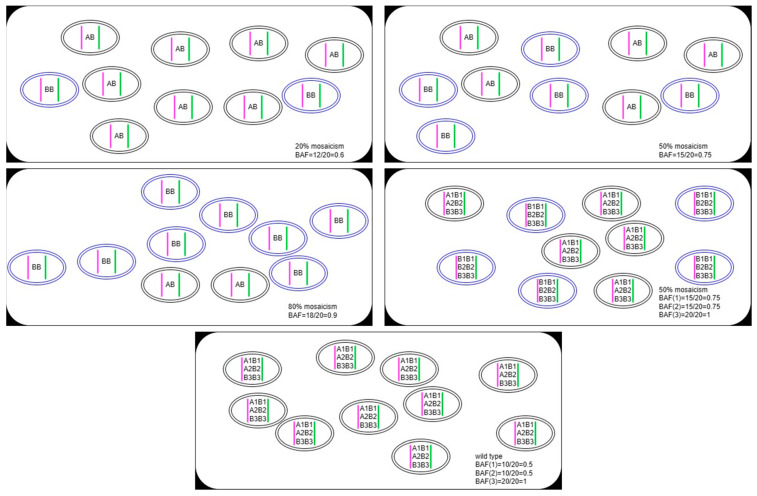
Schematic of change simulations of the B allele fraction (BAF) under different levels of mosaicism. Each ellipse represents a cell, for clarity, assuming each individual has 10 cells. The pink and green vertical lines represent a pair of homologs. The BAF value gradually increases as the mosaicism ratio rises. Comparing the scenario of 50% mosaicism with the wild type, no alteration is observed in the count of homozygous allele, indicating that homozygous alleles are helpless in aiding the recognition of mosaic AOH.

**Figure 2 diagnostics-13-02895-f002:**
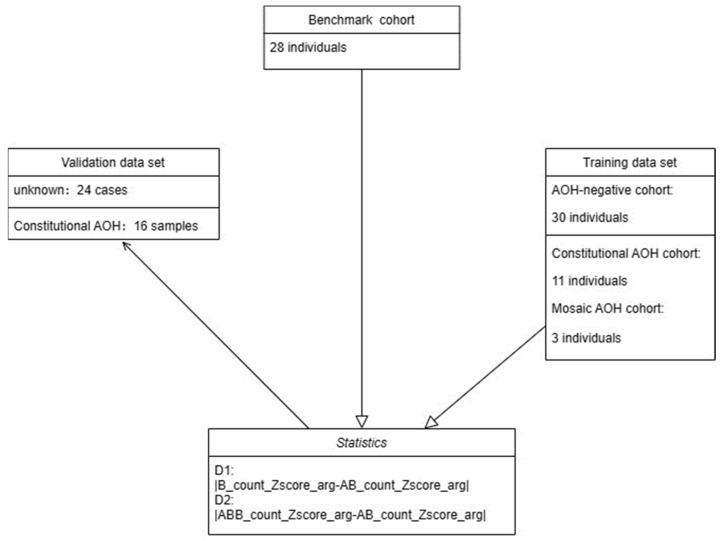
Development and validation of the CMA-seq algorithm for mosaic AOH detection.

**Figure 3 diagnostics-13-02895-f003:**
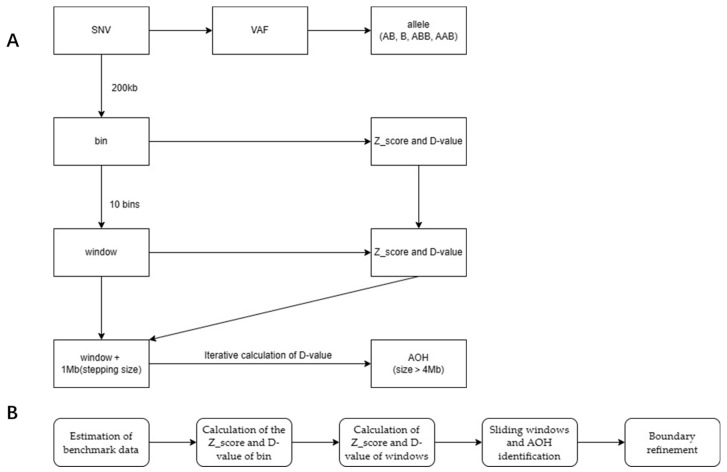
Algorithm and procedure of the bioinformatics analysis. (**A**) Showed the analysis algorithm converting SNV signals to AOH identification. (**B**) Showed the calculation and identification steps.

**Figure 4 diagnostics-13-02895-f004:**
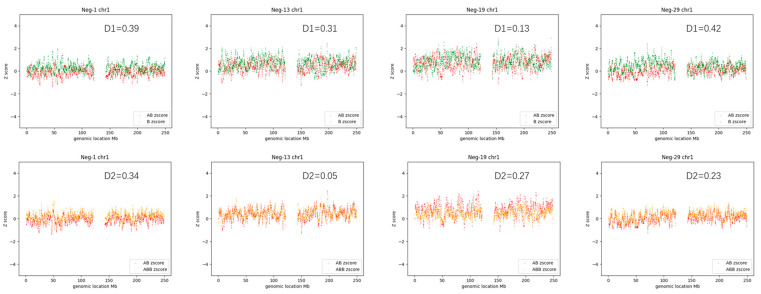
Z score average dot-plot of chromosome 1 of four randomly selected AOH-negative samples (Neg-1, Neg-13, Neg-19, and Neg-29). The red curve indicates the AB_count_Zscore dot-plot, the green curve indicates the B_count_Zscore dot-plot and the orange curve indicates the ABB_count_Zscore dot-plot. The D-value is displayed in the upper right corner. The upper four graphs were named the ‘D1 plot’, and the lower four graphs were named the ‘D2 plot’ hereafter.

**Figure 5 diagnostics-13-02895-f005:**
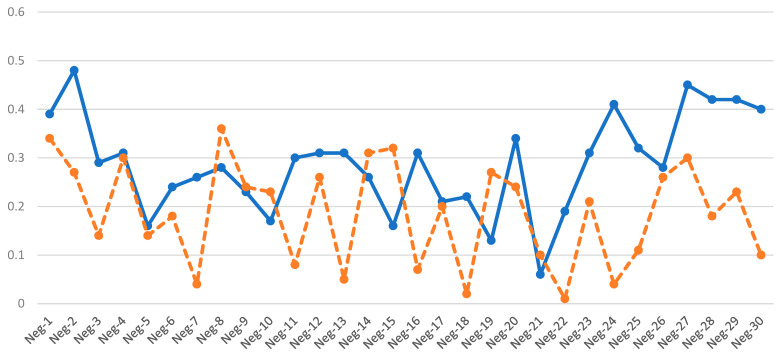
D1 and D2 values of chromosome 1 of 30 AOH-negative samples. All samples met the criteria of D1 < 0.5 (blue) and D2 < 0.6 (orange).

**Figure 6 diagnostics-13-02895-f006:**
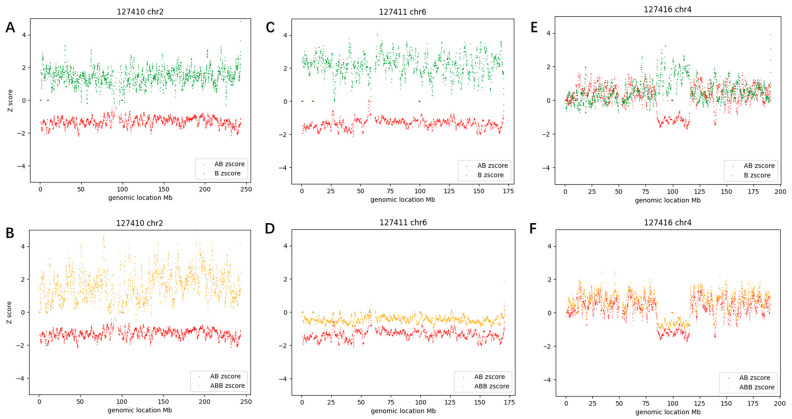
The D1 plot and D2 plot of three constitutional AOH cases. (**A**,**B**) showed the results of case #59, (**C**,**D**) showed the results of case #58, and (**E**,**F**) showed the results of case #53. Note a dramatic ‘gap’ formed within the curves of AB count_Zscore (red) and B count_Zscore (green) in all constitutional AOH regions, implying the existence of a contiguous stretch of homozygous variants.

**Figure 7 diagnostics-13-02895-f007:**
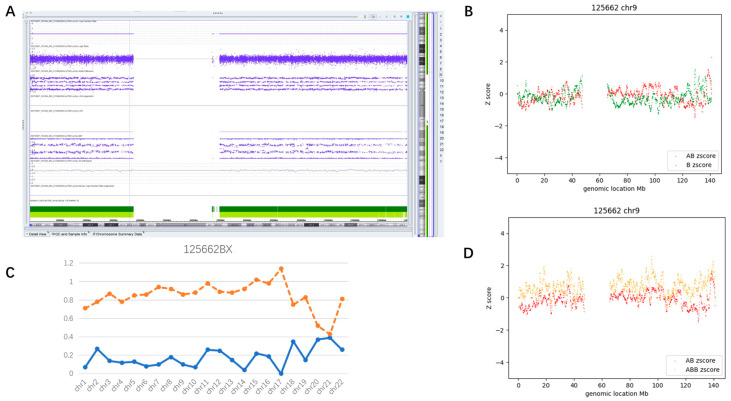
Example of whole genome mosaic AOH. (**A**) Showed the results of CMA (chr9 as a representative); (**B**,**D**) showed the D1 plot and D2 plot of chr9; (**C**) showed the D1 (orange) and D2 (blue) values of all autosomes.

**Figure 8 diagnostics-13-02895-f008:**
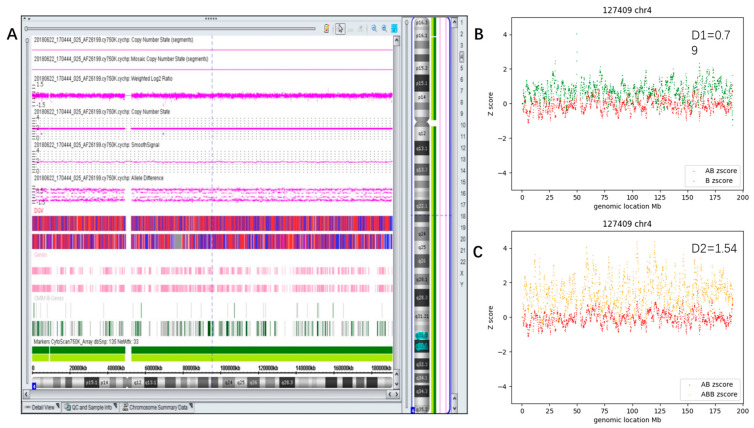
Example of whole chromosome mosaic AOH. (**A**) Showed the results of CMA; (**B**,**C**) showed the D1 plot and D2 plot of chr4. The D-value is displayed in the upper right corner.

**Figure 9 diagnostics-13-02895-f009:**
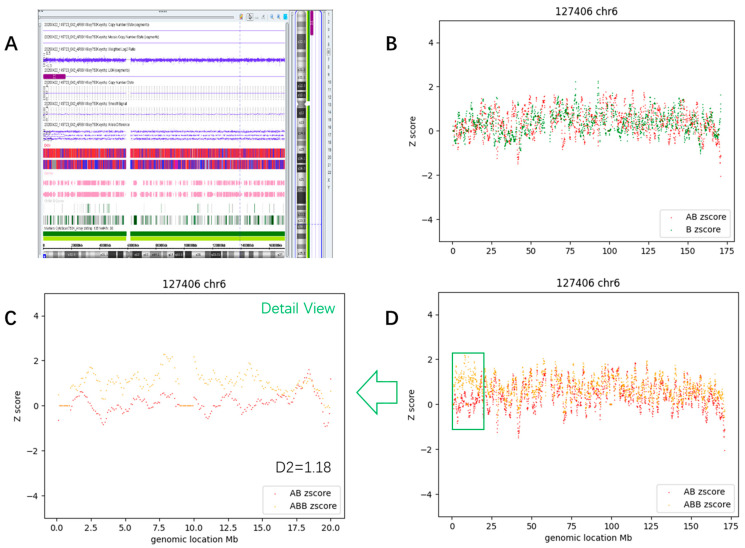
Example of mosaic segmental AOH. (**A**) Showed the results of CMA; (**B**,**D**) showed the D1 plot and D2 plot of chr6. (**C**) showed a detailed view of the tiny AOH region.

**Figure 10 diagnostics-13-02895-f010:**
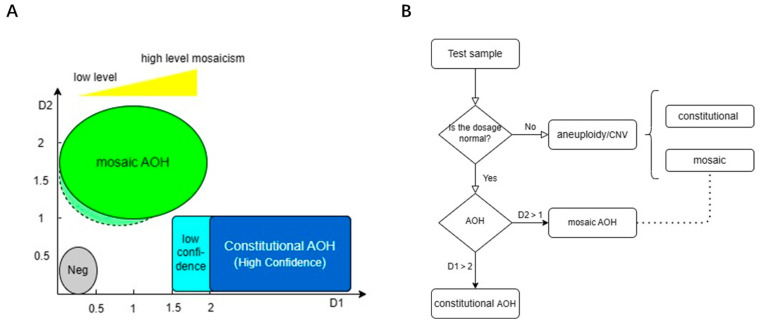
Thresholds for determining mosaic and constitutional AOH and detection pipeline. (**A**) Delineated the zone of AOH, light color indicates loose cutoffs, dark color indicates stringent cutoffs, and Neg represents AOH-negative. The yellow triangle indicates the mosaicism level. (**B**) Showed the detection pipeline of AOH and aneuploidy/CNV, and the dashed line indicates the possibility of co-occurrence of mosaic aneuploidy/CNV and AOH.

**Figure 11 diagnostics-13-02895-f011:**
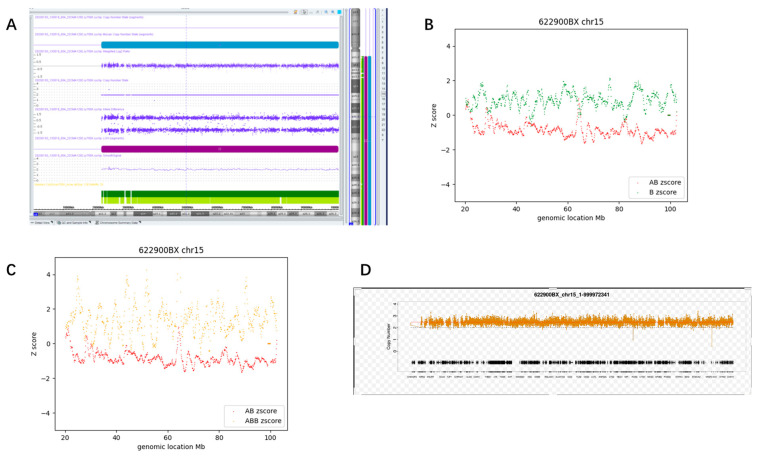
New mosaic AOH case identified by our detection algorithm using the abortive placenta. (**A**) Showed the results from CMA; (**B**,**C**) showed the D1 plot and D2 plot of chr15; (**D**) showed the dosage level of chr15 (×2.47).

**Table 1 diagnostics-13-02895-t001:** Thresholds for determining mosaic and constitutional AOH.

Category	Subgroup	D1	D2
Negative		<0.5	<0.6
AOH (mosaic)	Whole chromosome	<1.5	>1
	Segmental (size > 10 Mb)	<1.5	>1
	Whole genome	<0.5 *	>0.85 *
AOH (constitutional)	Whole chromosome	>2 (stringent); >1.5 (loose)	<1
	Segmental (size > 4 Mb)	>2 (stringent); >1.5 (loose)	<1
	Whole genome	>2 *	>2 *

Note: * represents inconclusive thresholds and requires more training.

**Table 2 diagnostics-13-02895-t002:** Comparison of AOH results of 24 validation cases by CMA and CMA-seq.

Case Code	CMA	Size (Mb)	CMA-Seq	Size (Mb)
#164	arr[hg19] 1p35.2p33(31,316,029_47,996,204) ×2 hmz	16.7	pos-chr1:23,000,000–49,000,000	26
			pos-chr12:69,000,000–78,000,000	9
#165	arr[hg19] 15q11.2q26.3(22,817,871_102,397,317) ×2 hmz	79.6	pos-chr15:0–103,000,000	Whole chr
#166	arr[hg19] 10p13p11.21(13,817,396_36,384,851) ×2 hmz	22.6	pos-chr10:12,000,000–37,000,000	25
			pos-chr8:33,000,000–58,000,000	25
#168	arr[hg19] 18q21.31q23(55,513,044_77,997,606) ×2 hmz	22.5	pos-chr18:55,000,000–78,000,000	23
#170	arr[hg19] 12q15q21.31(70,806,679_83,527,548) ×2 hmz	12.7		
#171	arr[hg19] 4p16.2p15.32(5,771,557_15,985,454) ×2 hmz	10.3	pos-chr4:8,000,000–16,000,000	8
	arr[hg19] 6p22.3p21.1(15,751,330_41,796,817) ×2 hmz	26	pos-chr6:14,000,000–40,000,000	26
#172	arr[hg19] 22q11.1q13.33(16,888,900_51,157,531) ×2 hmz	34.3	pos-chr22:0–52,000,000	Whole chr
#173	arr[hg19] 1q23.3q25.3(161,955,758_180,706,775) ×2 hmz	18.7	pos-chr1:160,000,000–185,000,000	25
#167	arr[hg19] 15q11.2q26.3(22,817,871_102,397,317) ×2–3 hmz	79.6	pos-chr15:0–103,000,000	Whole chr

## Data Availability

The datasets used and/or analyzed during the current study are available from the corresponding author upon reasonable request.

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
