# Peer review of "Detection of Mosaic Absence of Heterozygosity (AOH) Using Low-Pass Whole Genome Sequencing in Prenatal Diagnosis: A Preliminary Report"

_diagnostics, 2023, doi:10.3390/diagnostics13182895_

Round 1
Reviewer 1 Report
The title of the manuscript as well as the introduction is very confusing both in terminology and written language.
Titled “Detection of Mosaic Absence of Heterozygosity (AOH) Using Chromosome Analysis by Medium Coverage Whole Genome Sequencing (CMA-seq) in Prenatal Diagnosis” By Yan Lü et.al.
Lü et al. apparently tried to develop a bioinformatic algorithm using medium depth (5-8x) genome sequencing to detect mosaicism by way of B allele frequency.
The authors validated the algorithm on a limited number of positive cases and showed the ability to detect both constitutional abnormalities and mosaicism. However, the sensitivity and specificity of detecting mosaicism requires additional validation.
This reviewer found the title and use of terminology in the manuscript very confusing. For example, in the title: [1] chromosome analysis was used to study chromosomes, there were no data demonstrate the use of chromosome. [2] CMA-seq, CMA represents chromosome microarray analysis, which is using array-based technology, not sequence technology [3] Mos AOH, I cannot find any literature by this name.
This reviewer recommends revision of the manuscript into a language that can communicate their work to the reader of the journal. The citations used in the manuscript gave plenty examples how to use the established terminology for revision.
Comments on the developing algorithm for detection of mosaicism using B allele frequency observed from the medium read depth genome sequencing:
P2 The authors classify all SNPs into four types of genotypes (AB, B, ABB, AAB). I understand that AA is not used because they are not called on the VCFs as variants. The authors should better explain how they select SNPs across the genome that are used in the determination of AOH status. The genotype B is representing a pure B signal, but the genotype could also be represented as BB/BBB to be more accurate with the actual alleles.
P5 figure 3. Described in 2.3.3.2, if the Z score is calculated with the count of allele genotypes within 200kb bins, the arrow between the box “allele (AB, B, ABB, ABB) and Z Score and δ” should not exist.
P6. 2.3.2: The four categories of genotype classification should be mutually exclusive. Would ABB better have represented as 0.65>VAF>0.99?
P6. 2.3.3: the authors could elaborate more on the logic behind the calculation in addition to the description of the calculations that is present. The authors do mention the rationale behind introducing δ2 in the discussion, but they can also introduce it in the methods section.
Please match the case numbers in the figures and text.
Page 8 3.3.1: Was maternal cell contamination ruled out in case 28?
Since the paper focuses on prenatal application of this algorithm, the authors can also commentate on the utility of this tool in the detection of maternal cell contamination.
The authors should commentate on how many AOH calls are typically called by the algorithm in typical negative cases and whole chromosome AOH cases? How many calls require manual review. This will provide insight to the specificity of the caller.
Although this may be obvious, the authors should mention that this method cannot detect uniparental heterodisomy on a proband-only analysis.
It would be helpful to have someone in the field fluent in English language to revise the manuscript
Author Response
Dear Editor and Reviewer
On behalf of my co-authors, we thank you very much for giving us an opportunity to revise our manuscript, and we also appreciate reviewers very much for their positive and constructive comments and suggestions on our manuscript entitled “Detection of Mosaic Absence of Heterozygosity (AOH) Using Medium Coverage Whole Genome Sequencing in Prenatal Diagnosis” (Manuscript ID: diagnostics-2555688).
We revised the manuscript according to these comments and suggestions. In general, we have tried our best to revise our manuscript and provide the point-by-point responses. All changes were marked in red using the “Track Changes” function in the revised manuscript.
Please see below, in blue, for a point-by-point response to the reviewers’ comments and concerns.
- The authors validated the algorithm on a limited number of positive cases and showed the ability to detect both constitutional abnormalities and mosaicism. However, the sensitivity and specificity of detecting mosaicism requires additional validation.
On the basis of the original 24+10 validation samples, we add six constitutional AOH validation samples. In addition, the clinical sensitivity and specificity was calculated by the 24 new samples, see the end of 3.5 section for detail. However, due to scarce of mosaic AOH samples available, we cannot convincingly calculate the sensitivity and specificity of detecting mosaicism.
2.This reviewer found the title and use of terminology in the manuscript very confusing. For example, in the title: [1] chromosome analysis was used to study chromosomes, there were no data demonstrate the use of chromosome.
According to your suggestion, “Chromosome Analysis By” has been removed from the title.
- [2] CMA-seq, CMA represents chromosome microarray analysis, which is using array-based technology, not sequence technology
CMA-seq is a product name, it is an abbreviation for “Chromosome Analysis by Sequencing”. According to your suggestion, “CMA-seq” has been removed from the title to avoid confusion, but it is retained in the main text and provide an explanation for the terminology.
- [3] Mos AOH, I cannot find any literature by this name.
The bad spelling appears in the initial draft and has been corrected.
- The authors should better explain how they select SNPs across the genome that are used in the determination of AOH status.
We explained it from three aspects. First, any SNVs with read depth<5, or>(arg+4std)- fold were discarded, see 2.3.1 for detail; second, AAB alleles were not used for AOH identification due to they are greatly influenced by sequencing depth (Fig S1); third, AA is not used because they are not called on the VCFs (Variant call files).
- The genotype B is representing a pure B signal, but the genotype could also be represented as BB/BBB to be more accurate with the actual alleles.
We have corrected it according to your suggestion. BB is abbreviated as B in the following text. See 2.3.2 for detail.
- P5 figure 3. Described in 2.3.3.2, if the Z score is calculated with the count of allele genotypes within 200kb bins, the arrow between the box “allele (AB, B, ABB, ABB) and Z Score and δ” should not exist.
According to your suggestion, the arrow between the box “allele (AB, B, ABB, ABB) and Z Score and δ” has been removed.
- P6. 2.3.2: The four categories of genotype classification should be mutually exclusive. Would ABB better have represented as 0.65>VAF>0.99?
According to your suggestion, we have corrected the “ABB (VAF>0.65)” into “ABB (0.65<VAF<1)”.
- P6. 2.3.3: the authors could elaborate more on the logic behind the calculation in addition to the description of the calculations that is present. The authors do mention the rationale behind introducing δ2 in the discussion, but they can also introduce it in the methods section.
According to your suggestion, we further explained the logic in Fig1.
- Please match the case numbers in the figures and text.
The stupid mistakes have been corrected.
- Page 8 3.3.1: Was maternal cell contamination ruled out in case 28? Since the paper focuses on prenatal application of this algorithm, the authors can also commentate on the utility of this tool in the detection of maternal cell contamination.
We did not conduct a QF-PCR test to exclude the possibility of MCC as the parents’ genetic information is not available, however, the CMA provides information on SNPs that allows us to identify the existence of MCC by analyzing allele differences. In the Quality control of NGS data, the possibility of MCC can be ruled out by VerifyBamID2 (2.3.1).
12.The authors should commentate on how many AOH calls are typically called by the algorithm in typical negative cases and whole chromosome AOH cases? How many calls require manual review. This will provide insight to the specificity of the caller.
No AOH region, whether constitutional or mosaic, was called in the 30 negative cases. Every AOH call needs manual adjustment due to AOH boundary need refinement (2.3.3.5). In the actual operation, we will use δ1 plot and δ2 plot to accelerate the completion of boundary optimization.
13.Although this may be obvious, the authors should mention that this method cannot detect uniparental heterodisomy on a proband-only analysis.
The relevant description has been added to the Discussion section.
Once again, thank you very much for your comments and suggestions. And we hope that the revised manuscript can be accepted by Diagnostics. If further revision is necessary, please contact me at: chenlu.xu@amcarelab.com.cn
Thank you and best regards.
Sincerely yours,
Chenlu Xu, Yan Lü, Victor Wei Zhang and Qingwei Qi

Reviewer 2 Report
Review:
This article focuses on the detection of mosaic loss of heterozygosity (LOH) using genomics methods. It discusses the differences between mosaicism and chimerism and highlights the importance of identifying mosaic LOH in clinical settings. The study presents a detailed methodology involving genotype status determination and statistical analysis to classify LOH regions. This article also talks about a new bioinformatic algorithm that can detect mosaic AOH and optimized the existing method for constitutional AOH identification. The results demonstrate differentiation between negative, constitutional, and mosaic LOH cases. The article concludes by emphasizing the clinical relevance of the findings and potential applications in genetic diagnostics.
The study's comprehensive approach to detecting mosaic loss of heterozygosity (LOH) is commendable, shedding light on the complexities of genetic variations and their clinical implications. However, as a reviewer, I have some comments that need to be addressed.
Minor Comments:
1. The detailed description of sample preparation is useful. It would be helpful to mention any quality control measures taken to ensure the accuracy of the obtained sequencing data (for example: Uniformity and duplication and coverage at 10x, 20x, or 50x.)
2. The approach of classifying SNVs into genotype statuses based on VAF is rational. However, it might be useful to discuss how the classification might be affected by factors like sequencing depth, read alignment accuracy, and heterogeneity in cell populations.
3. It might be valuable to explain how you determined the optimal bin size of 200 kb for calculating statistics and why this choice was made.
4. The establishment of criteria for negative samples is crucial. However, consider explaining the rationale behind the specific thresholds (δ1 < 0.5, δ2 < 0.6) that were selected.
5. Could you please elaborate the calculation of δ1 and δ2 and what was the rationale behind the calculation of δ2 with fornula “δ2=| ABB_count_zscore_arg-AB_count_zscore_arg” when you already have B_count_zscore_arg
6. Could you please elaborate the arg+4sd ?
7. Please consider comparing your results with existing literature on AOH detection to highlight your method's innovation or comparison with other well-known tools in markets such as Nxcinical, Fabric or other tools etc.
Author Response
Dear Editor and Reviewer,
On behalf of my co-authors, we thank you very much for giving us an opportunity to revise our manuscript, and we also appreciate reviewers very much for their positive and constructive comments and suggestions on our manuscript entitled “Detection of Mosaic Absence of Heterozygosity (AOH) Using Medium Coverage Whole Genome Sequencing in Prenatal Diagnosis” (Manuscript ID: diagnostics-2555688).
We revised the manuscript according to these comments and suggestions. In general, we have tried our best to revise our manuscript and provide the point-by-point responses. All changes were marked in red using the “Track Changes” function in the revised manuscript.
Please see below, in blue, for a point-by-point response to the reviewers’ comments and concerns.
- The detailed description of sample preparation is useful. It would be helpful to mention any quality control measures taken to ensure the accuracy of the obtained sequencing data (for example: Uniformity and duplication and coverage at 10x, 20x, or 50x.)
The detailed description of sample preparation and quality control have been added, see 2.3.1 for detail.
- The approach of classifying SNVs into genotype statuses based on VAF is rational. However, it might be useful to discuss how the classification might be affected by factors like sequencing depth, read alignment accuracy, and heterogeneity in cell populations.
We removed loci with a depth higher than (avg+4std)-fold, due to they are likely located in the high CG or repetitive region of the genome, mutation detection of these regions is inaccurate. The influence of sequencing depth on genotyping can be found in Supplementary Fig S1. The effect of different levels of mosaicism on genotyping can be found in Fig1.
- It might be valuable to explain how you determined the optimal bin size of 200 kb for calculating statistics and why this choice was made.
Add the following content:The cause for selecting 200kb as bin size can be found in reference [17].
- The establishment of criteria for negative samples is crucial. However, consider explaining the rationale behind the specific thresholds (δ1 < 0.5, δ2 < 0.6) that were selected.
The essence of δ is the D-value of Z score. The Z score reflects the degree of deviation between the testing sample and normal individuals. If the deviation of the testing sample is not significant, reflecting as the low δ value, then this sample can be a negative sample.
- Could you please elaborate the calculation of δ1 and δ2 and what was the rationale behind the calculation of δ2 with formula “δ2=| ABB_count_zscore_arg-AB_count_zscore_arg” when you already have B_count_zscore_arg
Homozygous alleles were not used to identify mosaic AOH. The rationale behind the calculation of δ2 is the seesaw effect in the count of ABB alleles and AB alleles.
- Could you please elaborate the arg+4sd ?
We have used “arg+4std (mean value plus four standard deviation)” as you suggested.
- Please consider comparing your results with existing literature on AOH detection to highlight your method's innovation or comparison with other well-known tools in markets such as Nxcinical, Fabric or other tools etc.
We have compared our method with other representative methods using 45 clinical cases. The results of CMA-seq are similar to or better than comparable methods but the comparison results will be reflected in another paper being written.
Once again, thank you very much for your comments and suggestions. And we hope that the revised manuscript can be accepted by Diagnostics. If further revision is necessary, please contact me at: chenlu.xu@amcarelab.com.cn
Thank you and best regards.
Sincerely yours,
Chenlu Xu, Yan Lü, Victor Wei Zhang and Qingwei Qi

Round 2
Reviewer 1 Report
please see attached for my comments.

Quality of English Language need improvement. see attached for my comments.
